# Evaluation of Farm Fresh Food Boxes: A Hybrid Alternative Food Network Market Innovation

**Marilyn Sitaker [1],\*, Jane Kolodinsky [2] , Weiwei Wang [2], Lisa C. Chase [3], Julia Van Soelen Kim [4], Diane Smith [5] , Hans Estrin [3], Zoe Van Vlaanderen [2] and Lauren Greco [3]**

1 Ecological Agriculture and Food Systems, The Evergreen State College, Olympia, WA 98505, USA
2 Center for Rural Studies, University of Vermont, Burlington, VT 05405, USA; Jane.Kolodinsky@uvm.edu (J.K.); Wwang@uvm.edu (W.W.); zvanvlaa@uvm.edu (Z.V.V.); Lgreco@uvm.edu (L.G.)
3 Extension, University of Vermont, Brattleboro, VT 05301, USA; Lisa.Chase@uvm.edu (L.C.C.); Hans.estrin@uvm.edu (H.E.)
4 Cooperative Extension, University of California, Novato, CA 94947, USA; Jvansoelen@ucanr.edu
5 Extension, Washington State University, Burlington, WA 98233, USA; diane.smith@wsu.edu
\* Correspondence: msitaker@gmail.com; Tel.: +1-206-395-7501

**Abstract:** Using a mixed-methods design, we evaluated Farm Fresh Food Box (F3B) a market innovation designed to expand producer markets, stabilize rural retail businesses, and improve rural food access. In the F3B model, pre-ordered Community Supported Agriculture (CSA)-style produce boxes are sold through rural retail outlets. F3B was implemented from 2016 to 2018 as part of a United States Department of Agriculture (USDA)-funded multi-state extension and research collaboration project in 3 geographically diverse and rural areas: Vermont, Washington, and California. The F3B evaluation aimed to (1) assess market potential; (2) determine logistics for successful implementation; (3) describe the benefits and drawbacks for farmers and retailers; and (4) measure consumers' attitudes and purchase behavior. A national market survey indicated consumers would be likely to purchase F3B if it was perceived to offer good value on fresh local produce, without need for a subscription. The model put a few additional labor burdens on farmers and retailers, but required time for relationship-building and more record-keeping time for farmers. Those who purchased a F3B were generally satisfied with the quality, quantity and variety of produce each week and a high proportion considered F3B to be a good value for the money. As a new business innovation, F3B showed only modest profit, but retailers and farmers felt it was worthwhile to expand their customer base, promote their brand and develop their partnership. F3B began a means to address flattened growth in direct to consumer produce sales, food deserts and dwindling retail options for fresh foods in rural areas. We discuss F3B as a potential solution to food system weaknesses exposed by the COIVD-19 pandemic because it offers touch-free, high-quality local produce ready for curbside pickup at a convenient location.

**Keywords:** sustainability; agribusiness; local foods; hybrid values-based supply chain; rural development; COVID-19

## 1. Introduction

The Farm Fresh Food Box (F3B) is an alternative food network (AFN) market innovation that combines features of direct to consumer (DTC) and values-based supply chain (VBSC) models in order to expand producer markets, stabilize rural retail businesses, and improve rural food access [1,2]. In the F3B model, pre-ordered CSA-style produce boxes are sold through rural retail outlets. F3B was

recently implemented from 2016 to 2018 as part of a USDA-funded multi-state extension and research collaboration project in 3 geographically diverse and rural areas: Vermont, Washington, and California.

F3B aims to address threats to the viability of small- and medium-sized farms and rural grocery stores resulting from the growth of large-scale farms, urban chain supermarkets, and big-box stores. Large centralized farms benefit from technological efficiencies and economies of scale, enabling them to outcompete smaller players [3,4]. It is increasingly difficult for small and mid-sized farmers to find conventional markets that provide a sufficient return, given the lower volumes these farmers are able to supply [5]. Rural retailers are challenged by distributors who require large-volume orders or refuse out-of-the way deliveries, and by retail regulations written with larger businesses in mind [6]. Additionally, a shift in rural shopping habits to urban centers puts rural economies at risk because rural grocery stores cannot match the scale and efficiency of big box stores that offer consumers a relatively cheap and efficient shopping experience [3,7–9].

The challenges rural retailers encounter have made it difficult for consumers to buy fresh, healthy produce in their own communities [10–17]. Research confirms that rural residents often have a limited choice of places to buy food in their own community [14,17–19]., Due to the relationship between intake of fresh, whole foods and risk of obesity and chronic diseases [6,20], diminished access to fresh and affordable produce significantly impacts the health of rural residents [10,14].

In response to these trends, small- and medium-sized farms have focused on selling through direct-to-consumer (DTC) venues like farmers markets and community supported agriculture (CSA), among others. These markets offer benefits like higher price points and local economic growth [4,21]. Furthermore, recent trends suggest saturation of these markets [1,2,21], prompting farmers to experiment with market innovations. F3B is one such strategy that aims to address the DTC markets and erosion of rural retailer space by shifting rural consumer food dollars away from large commercial centers and industrial agriculture and bring them back towards rural retailers and farms [2,22,23].

## 2. Literature Review

### 2.1. Trends in Direct Sales by Small and Medium Sized Farms

DTC marketing strategies emerged in response to the market dominance of industrial supply chains that favor large-scale farms and chain grocery stores, thus weakening the economic viability of smaller farms and rural economies [3,7–9,24]. DTC models are part of a larger effort to create alternative food networks (AFN) that relocalize the food system by featuring shorter supply chains and emphasizing the value of sustainable production methods, fair prices for farmers, and support for rural economies [21,25–28].

Strategies that allow farmers to sell directly to retailers, institutions and consumers (as in farmers' markets and CSAs) are an attractive way for farmers to generate profit while retaining control over price-setting and brand identity. Each model has its benefits and constraints. Selling at the farmers' market requires that producers take time away from farming to transport their product and staff their stall; at day's end, unsold produce may be wasted [29]. CSAs offer the advantage of decreasing the farmer's risk through an up-front subscription, while requiring relatively little time spent transporting shares to the drop off site [21], but studies have found that while CSA programs enable farmers to cover their operating expenses, they may not adequately cover labor costs [30,31].

The literature shows that DTC sales of locally produced foods can be profitable for producers [21,32–35]. Additionally, there is evidence that local economies benefit from direct sales through farmers markets [36–39], farm to school programs [40–43], and food hubs [44]. For instance, a "buy local" campaign in farmers markets in South Carolina generated $751,000 in sales, $104,000 in earned income, and 26.4 in full-time equivalent jobs [37].

Sales from DTC markets grew rapidly between 1992 to 2007 but started to flatten in 2012, likely as a result of waning consumer interest and market saturation [45]. In response, farmers have needed to become more innovative to retain an existing customer or attract new ones [46–48]. For example,

to accommodate and attract low-income customers, CSAs have experimented with subsidized or cost-offset CSA (CO-CSA) shares at reduced prices, payment plans, working shares, transportation assistance, and bartering [48–50].

Some farmers selling through DTC channels have shifted to also sell through intermediated channels such as restaurants and institutions, which in 2017 outperformed DTC sales 3:1 [4]. Direct intermediated sales may function as VBSC, in which farms, businesses and institutions engage in cooperative relationships. They rely on trust and communication to determine the division of labor [5,51], and are shaped by shared values, which might include goals beyond profit maximization [51–54]. In addition to creating new marketing channels for small and medium-sized growers with a potential price premium over commodity markets [33,55–57], these shorter VBSCs have potential to improve fresh food access to low-income communities by expanding beyond the geographic and cultural barriers of DTC.

Farmers can also sell through a food hub, which combines products from multiple farms so that individual and commercial buyers can benefit from lower cost and greater variety. Food Hubs can relieve farmers of the burdens of marketing, packing, distribution to multiple sites, and handling financial transactions [21,58,59]. However, the farm may not be amenable to selling through a food hub if the price point is below that of a farmer's market or CSA [51]. Further, meeting the demand for a large and specific quantity of fresh produce year-round at a specific price point can be challenging for smaller farms seeking to supply food hubs or sell to institutions directly [60].

### 2.2. Trends in Rural Retail

Historically, community stores were the centers for trade and social gathering places in rural communities. Modern life has challenged their viability. With larger centers of commerce accessible by car and many working away from their town of residence, consumers often shop elsewhere as part of their work commute [3,6]. Further, many rural consumers now choose to make a monthly shopping trip to a large chain grocery store in nearby urban centers, supplemented with smaller purchases [12,61]. These trends threaten the viability of rural grocery stores [62], as revenues shift from local to outside businesses. In contrast, money spent in the local area has an economic multiplier effect, helping to sustain an area's economy [63]. Independent grocery stores are also impacted by dollar stores and e-commerce that undercut prices and alter consumer shopping habits [64,65]. As a result, many small communities have lost their local grocers [66].

Rural community stores are challenged by the limitations in scale, as most distributors would prefer larger volume orders, particularly if they must deliver food to an out-of-the-way village center. Current retail regulations put compliance strain on small operations. Additionally, investigations of retailer concerns about stocking healthy food have revealed that retailers are challenged by lower consumer demand for fresh fruits and vegetables, lower profitability, lack of refrigerator space for highly perishable fresh produce, and inability to return unsold foods to suppliers [20,67–69].

Retailers who cannot overcome these barriers may be forced to limit their products, leaving rural residents without access to fresh produce in local stores and further driving consumers to shift their shopping dollars to larger centers of commerce outside the community. Yet solutions exist to overcome these challenges. For example, shopkeepers say they would be willing to increase their stock of healthy foods if there were monetary incentives to cover the extra cost of electricity to run coolers, subsidies for healthy foods, consumer education about healthy eating [20] as well as an increase in customer demand and a guarantee that fruits and vegetables would sell prior to ordering them [69].

### 2.3. Impact on Rural Consumers

As more small retailers limit their offerings or go out of business, rural consumers have found it harder to access a diverse array of healthy foods they can purchase locally [7,10,11,13–17], which impacts their consumption of fruits and vegetables [20,70–73]. The practice of travelling outside of rural communities to shop may contribute to eating less produce, when fewer trips are made to purchase

perishable fruit and vegetables [7,63,74]. Diminished access to fresh and affordable produce and the resulting effect on consumption of fresh, whole foods, can lead to obesity and chronic diseases, which are more prevalent in rural settings [69,71,75–77].

Innovative local food outlets can help redirect consumers away from centralized big box retailers back to their local communities, thereby keeping money in the local economy while making healthy, locally grown foods [21,78] available.

Many rural residents living in agricultural communities are not able to purchase the foods grown in their own communities [18,79]. Further, they may experience barriers when accessing local food: additional travel time required to pick-up local produce, prohibitive up-front costs of the pre-pay CSA model, perceived higher cost of local foods, the perceived elitism of shopping at a farmers market, or the very real affordability challenges of food [79].

### 2.4. Purpose and Advantages of F3B

F3B was developed to open new markets for farmers currently selling in a sluggish DTC market environment, improving upon limitations of the CSA model and incorporating advantages of VBSC. F3B was also intended to create a way for rural grocers to expand the variety and amount of produce offerings without investment in new equipment and space, without risk of waste from unsold items and spoilage. Finally, F3B aimed to improve consumer access to a variety of fresh, healthy local foods, without long term commitment or up-front expense, in convenient community locations along usual travel routes.

The F3B strategy has the potential to shift rural consumer food dollars away from distant, large-scale farms and urban chain supermarkets and bring them back towards local retailers and farms. To do this, F3B links small farms that typically sell through DTC venues with rural stores in a short VBSC. F3B features boxes of farmer-selected items that can be pre-ordered and picked up in neighboring rural retail stores. The retailer handles in-store advertising, orders, and payment transactions. Shifting from a direct to short VBSC model requires communication, coordination and shared decision making between farmer and retailer. It also requires that the unique embedded product values once conveyed to customers by the farmer alone must now be conveyed to the retailer who, in turn, is responsible for communicating it to the consumer. This model allows the farmer to retain ownership of the box and their products while selling through a more accessible venue for consumers [80]. The F3B addresses the previously discussed limitations of other DTC methods, including farm viability, retailer environment, and rural food access issues for consumers.

Figure 1 compares the costs of various models for consumers, farmers and retailers. Consumers find F3B costs on par with that of farm stands and grocery stores, but with lower cost and greater convenience than other DTC models. For farmers, F3B is lower-risk and more efficient than farmers' markets because boxes are pre-sold and delivered all at one time to one (or a few) sites. The retailer minimizes their profit margin on the F3B, gathering a small mark-up fee to cover transaction costs. However, the retailer may benefit from collateral sales of other items and a boost in customer traffic and loyalty. Thus, F3B fills a new market space that compares favorably or exceeds the benefit to consumers, farmers and retailers provided by other models.

### 2.5. Intervention Design and Setting

The F3B project was a tri-state collaboration of extension and research partners from the University of Vermont (UVM), Washington State University (WSU), Evergreen State College (TESC), and the University of California (UC). Local cooperative extension partners invited interested producers to participate, then reached out to recruit neighboring retailers. In the Spring of 2017, Washington and Vermont each worked with three farmer-retailer pairs implementing a full-season F3B pilot project (In 2017, Washington recruited a 4th retailer who dropped out before the end of the season. California was unable to participate in 2017 due to persistent wildfires in the region).

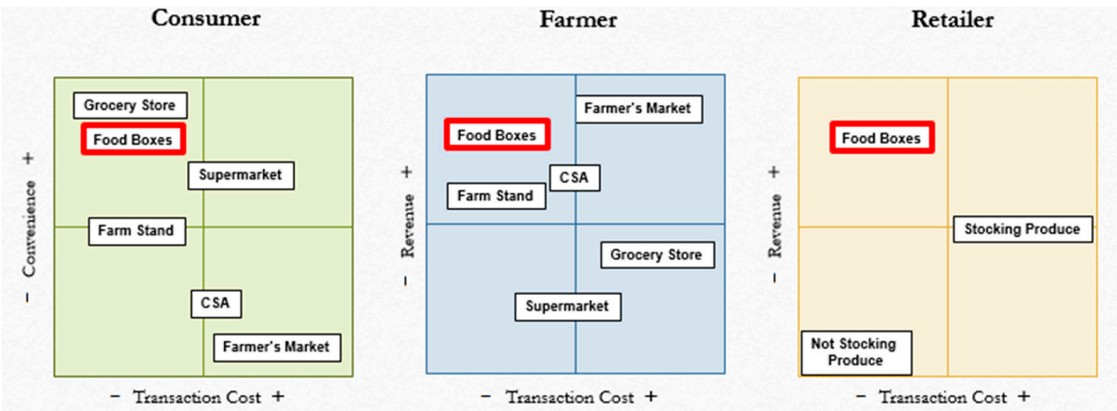

**Figure 1.** Cost Comparison of Various Models for Consumers, Farmers and Retailers.

In 2018, California recruited one farmer with three retail partners, Vermont replaced two farmer-retailer pairs from the previous season with two new pairs, while Washington maintained one farmer-retailer pair, and the other two farms found new retail partners. In all, there were nine farmers and 12 retailers across three states who completed at least one full season of F3B implementation. In general, farms were small and independently owned, and sold through at least one DTC market channel. Some farms also raised animals for meat and had some wholesale markets. In 2017, three retailers had gas stations at their stores, two were independent general stores, and one was a farm and feed store [80].

After enrollment, extension helped each farmer-retailer pair determine project logistics, provided tailored marketing materials, and gave ongoing technical support throughout the season.

F3B implementation varied in duration from one to six months (mean = 2.6 months). Farms set the box sizes and price, which ranged from $11 to $30. Two farmers offered both a full-size box and a smaller, lower-priced box. Specific logistical and marketing elements varied by location, community demographics, and store culture. In this paper, we describe characteristics for each state in aggregate form because describing individual farms and locations with more specificity would reveal the exact farm.

## 3. Study Design, Materials and Methods

The research aims of the F3B study were to:

1. Assess F3B market potential in 3 geographically diverse and rural areas
2. Determine strategies for recruitment, logistics, and marketing
3. Describe F3B benefits and challenges for farmers and retailers
4. Measure consumer response in terms of attitudes and purchase behavior

We used a mixed-methods design to assess each aim, based on quantitative and qualitative measures from primary and secondary data sources. In the following subsections, we describe the constructs, variables, data sources and analytic approach used to assess each aim. Table 1 provides a summary of primary and secondary data sources used.

**Table 1.** Research Aims and Data Sources.

| | Research Aim | F3B Marketing Survey | USDA/US Census | Farmer-Retailer Interviews | Purchaser Survey | Extension Notes and Records |
|---|---|---|---|---|---|---|
| 1. | Market potential | x | x | x | | |
| 2. | Recruitment, logistics, and marketing | | | | | x |
| 3. | Benefits and challenges | | | x | x | |
| 4. | Consumer response | | | | x | |

### 3.1. Aim 1: Assess F3B Market Potential

To assess F3B market potential, we used U.S. Census data at the block group level to document the commercial environment in terms of area demographics, transportation options, food access and potential competition (proximal DTC venues and conventional food retail outlets). Specific variables included total population, median age and income; distribution by race/ethnicity; and percentages of those with high school degree, living in poverty, receiving Supplemental Nutrition Assistance Program (SNAP) benefits, and driving cars to work [22]. Descriptive statistics were generated for the demographic, transportation, and geospatial variables, including counts, means, and proportions, which were then aggregated to the state level. We averaged statistics on retailer's locations in each state to facilitate comparisons across states.

We conducted a national market research survey to assess consumer demand in terms of regional attitudes towards local foods, utilization of existing DTC venues and interest in F3B. The national market research survey was a random-digit dial telephone survey of 399 U.S. adults (response rate = 12.3%), conducted by the University of Vermont Center for Rural Studies in March 2018 to inform F3B marketing efforts. The survey tool, developed by the research team, contained items on consumers' use of conventional and DTC market venues, attitudes towards locally grown foods, frequency and reasons for buying it, opinions about the F3B model and likelihood of purchase. Relative to all US adults in 2018, survey respondents had a higher proportion that identified as white (90% compared to 63%), male (63.8% compared to 48.7%), college-educated (53% compared to 35%), and older age (median 67, compared to 46.4). The research team generated simple descriptive statistics for the entire sample and for each of four regions (Northeast, South, Midwest and West) [81].

We included descriptions of farmer and retailer perceptions of how F3B might benefit their business and community, given their current clientele and foods offered by competitors, using data from the 2017 and 2018 post-season interviews conducted with participating farmers and retailers (data collection and analysis described under Section 3.3, below).

### 3.2. Aim 2: Determine Strategies for Recruitment, Logistics, and Marketing

Determination of strategies for recruitment and logistics were based on the experience of extension as the worked across diverse settings in three states. We used content analysis of the extension team's field notes and documented observations to identify strategies for recruitment and logistical arrangements that applied to all implementation sites, recognizing that each site would tailor these to fit the local community context and the needs of their business.

### 3.3. Aim 3: Describe F3B Benefits and Challenges for Farmers and Retailers

Extension records on box sales at each implementation site were used to describe F3B's profitability for farmers. We further assessed perceptions of F3B's benefits and challenges through post-season interviews with nine participating F3B farmers and 12 retailers, conducted in 2017 and 2018. Interview topics included motivations for participation, perceptions regarding the commercial environment for local foods, and perceived benefits and challenges of implementing the F3B model.

Interviews were conducted over the phone by members of the research team, using an interview guide developed by the research and extension teams. They were then recorded, transcribed verbatim by a third-party contractor into MS Word. De-identified transcripts were structurally coded in NVivo according to the interview guide, and using a codebook based on themes in the value-chain framework and the stacked beliefs framework [80].

### 3.4. Aim 4: Measure Consumer Response in Terms of Attitudes and Purchase Behavior

Assessment of consumer response was accomplished through analysis of data collected from purchasers, using a tool developed by the research and extension teams. A paper purchaser survey was included in each of the 607 F3B units sold during the 2017 and 2018 seasons, along with a stamped return envelope addressed to the local extension office. Fifty-nine customers returned the survey either by mail or in person at the store (response rate = 9.7%). Surveys were then forwarded to the F3B research team by the extension partner. The survey used short answer, Likert scale, and open-ended questions to ascertain purchasers' feedback on the affordability of F3B, as well Diffusion of Innovation characteristics such as relative advantage, trialability, complexity, compatibility, and observability. Descriptive statistics were generated for quantitative responses using both continuous and transformed categorical variables. Open-ended questions were coded based on major themes, using a codebook developed collaboratively by two researchers [23].

Recognizing that it would be valuable to understand why customers did not choose to purchase a F3B, extension partners made an attempt to conduct retail intercept surveys at some participating retail sites. However, this approach was unsuccessful and did not produce any usable data.

## 4. Results

### 4.1. Aim 1: F3B Market Potential

#### 4.1.1. Demographic Profile; Geo-Spatial Analysis

As shown in Table 2, census tracts in F3B implementation sites had small populations, though VT F3B retailers were generally located in more populous than their WA counterparts (average population 1750 and 740, respectively). As previously reported [22], most F3B communities had higher levels of education and income and lower levels of poverty and racial diversity than their state average, with the exception of a WA community with 26% non-white population. According to Rural Urban Commuting Area (RUCA) classifications, VT retailers were located in rural, micropolitan areas, or large urban centers, while WA retailers were in metropolitan commuter areas outside a nearby, larger city [22].

Measures of the transportation environment showed that F3B pickup locations were mainly in highly car-dependent areas, where between 83.6 and 88 percent of workers drove cars to work, on average (Table 2). Public transportation was mostly nonexistent. The ease with which customers could get to the retail store on foot was generally low, though walkability scores showed wide variation within states.

To address potential transportation barriers, farmers made a point of selecting retail partners situated along known customer travel routes [22]. Since most F3B farms were located fairly close to retail sites, farmers did not have to go very far to deliver boxes each week. In terms of potential competition, most F3B retailers had a competing supermarket (SM) within 3 miles, with the exception of one WA F3B retailer for whom the closest supermarket was 9.4 miles away. Competition from other DTC venues appeared to be sparse in most areas, as the majority of retail sites had no FM within a two-mile radius, and the driving distance to the closest FM was more than 15 miles for three sites [22].

**Table 2.** Demographic Profile of F3B implementation Sites.

| | Vermont | | Washington | |
|---|---|---|---|---|
| | **Range** | **Mean (SD)** | **Range** | **Mean (SD)** |
| **Demographics** | | | | |
| Population | 1057–2321 | 1750 (641) | 461–952 | 740(252) |
| Median Age | 38–47.4 | 42.5 (4.7) | 37.5–50.7 | 45.8 (7.2) |
| Median HH Income | $50,662–$93,281 | $69,949 ($21,595) | $24,830–100,735 | $66,438 ($38,477) |
| % H.S. Graduates | 10.0–20.0 | 14 (5.3) | 12.0–22.0 | 16 (5.3) |
| % Minority population | 3.0–6.0 | 5.0 (1.7) | 5.0–26.0 | 12.7 (11.6) |
| % Living in Poverty | 2.7–5.4 | 4.2 (1.4) | 0.0–12.0 | 4.8 (6.3) |
| % HH receiving SNAP | 3.0–25.0 | 12.3 (11.4) | 3–40.0 | 14.3 (22.3) |
| **Transportation Environment** | | | | |
| Walk Score | 24–44 | 34.3 (10.0) | 12–20 | 17.0 (4) |
| Transit Score | 0 | 0 | 0 | 0 |
| % Drive to work | 47.1–89.6 | 88.0 (5.2) | 70.6–93.3 | 83.6 (11.7) |
| **Geospatial characteristics** | | | | |
| F3B Farm-to-Store | 2.1–4.2 miles | 3.5 (1.2) | 0.9–9.9 miles | 5.5 (4.5) |
| # SM within a 2 mi radius | 0–2 | 0.7 (1.2) | 0–3 | 1.0 (1.7) |
| F3B Store to Supermarket | 0.2–6 miles | 2.3 (3.2) | 4.1–18.3 miles | 9.4 (7.8) |
| # FM within a 2 mi radius | 0–1 | 0.3 (0.6) | 0 | 0 |
| F3B Store to Farmers Market | 0–38 miles | 18.5 (19) | 4.2–19.4 miles | 9.8 (8.4) |

### 4.1.2. Contextual Factors, According to Farmers

Post-season interviews revealed that farmers were keenly aware of the impact of national trends in market saturation on local DTC sales, with many stating this motivated them to participate in the F3B project. They said their sales had been affected by the growing popularity of online grocery stores and meal kits. Farmers also noted the burgeoning availability of locally grown produce in neighboring conventional and alternative food outlets, which sometimes put them at a price disadvantage:

"Our food is going to be more expensive than at Costco or Walmart, I think the local agricultural products, in general, are really available now in almost all retail settings, you know, stores, restaurants, a lot of CSAs, a lot of farm stands, a lot of farmer's markets." Farm VT2, 2017

Some farmers noted that geography sometimes advantaged F3B. One remote farm (WA3) said their CSA membership included well-educated staff in a nearby national park who were selective about the quality and taste of their food yet had few nearby sources other than her farm. This farmer felt that geographic isolation created a niche for F3B in convenience stores/gas stations. Yet even within generally wealthier areas, there were pockets of need. Farm WA3 commented that an economic boom in the nearby urban center had pushed low-wage workers to shop further from home as they searched for more affordable foods, and affirmed that long distances between stores made it harder for low-income people without cars to get food.

In post-season interviews, farmers observed that different types of stores attracted customers with different expectations and purchase objectives. Customers shopping at a country store might expect to see locally grown foods sold there, and want to buy them. Further, these customers might have more confidence in new products the store owner introduced:

"It's almost like people think, 'Oh, this is worthwhile, because, you know, the [store owners] are into it.'" Farm VT1, 2017

Conversely, convenience store shoppers would be less likely to expect to find fresh produce in these stores:

"You go grab something quick, you grab your lunch quick, or you get that gallon of milk that you forgot, or maybe they have ice cream on sale, but you're not going out to go grocery shopping when you stop there." Farm VT2, 2017

### 4.1.3. Consumer Attitudes

We conducted the F3B National Marketing survey to assess consumers' previous experience with DTC venues and buying local produce, as well as their reactions to the F3B model. While a high percentage of US consumers have purchased food through a DTC venue sometime in their life (79.1%), untapped markets may exist in the Western region, particularly in farmers market and farm stand venues. This aligns with lifetime experience with DTC shopping documented elsewhere [82]. Figure 2 shows that lifetime prevalence varies by region, being highest in the Northeast (89.7%) and lowest in the West (64%). Echoing other studies, respondents' top motivations for buying locally grown foods were freshness and taste (93%) wholesomeness (86%) and a desire to support local farmers (77%) and local economies (75%). Among those who had purchased locally grown items, the biggest barriers to buying local foods were an inability to get specific items when they wanted them (56%) and perishability (40%); only 20% mentioned the time it takes to clean, chop and store as an impediment. For those who had never purchased locally grown produce, 49% said seasonality was the top reason not to purchase it, 42% said it was too expensive and 41% felt the hours of operation for DTC venues were inconvenient.

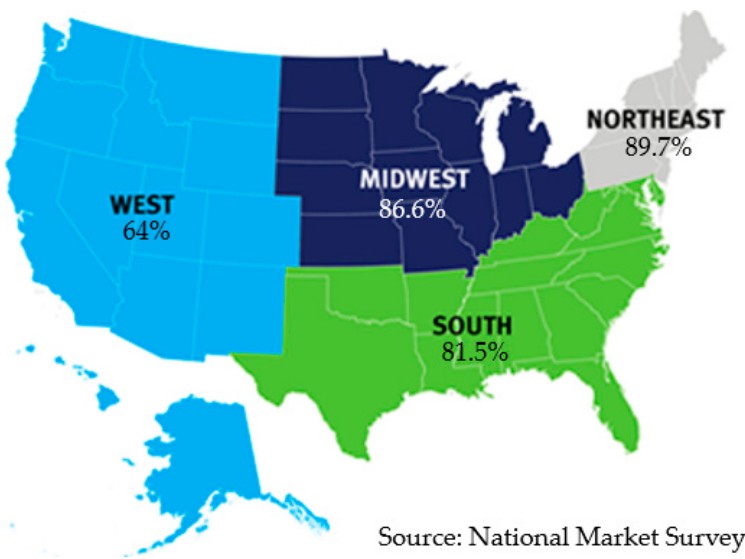

**Figure 2.** Lifetime prevalence of purchasing from any DTC venue.

When presented with a description of F3B, and how it would work, 74 percent of national market survey respondents said being able to get fresh seasonal produce would influence their decision whether or not to purchase one. Other influential factors included concerns about getting too much of some items and too little of others (67%); constraints on pick-up times (63%) and the cost compared to grocery stores (61%). Factors less frequently cited included only getting farmer-selected items (49%), getting unfamiliar vegetables (46%) and the additional time required to prepare raw foods (34%).

As shown in Figure 3, respondents said that they would be more likely to purchase a F3B if the quality of its contents justified the extra cost, closely followed by no weekly commitment to buy it. Other things that increased the likelihood of purchase included getting items that appeared fresher than similar items in the supermarket, getting a variety of produce for a flat price, and knowing someone who had already tried F3B (observability). Getting recipes, tasting a recipe featuring items in that week's box, and inclusion of lightly processed produce also increased the likelihood of purchase.

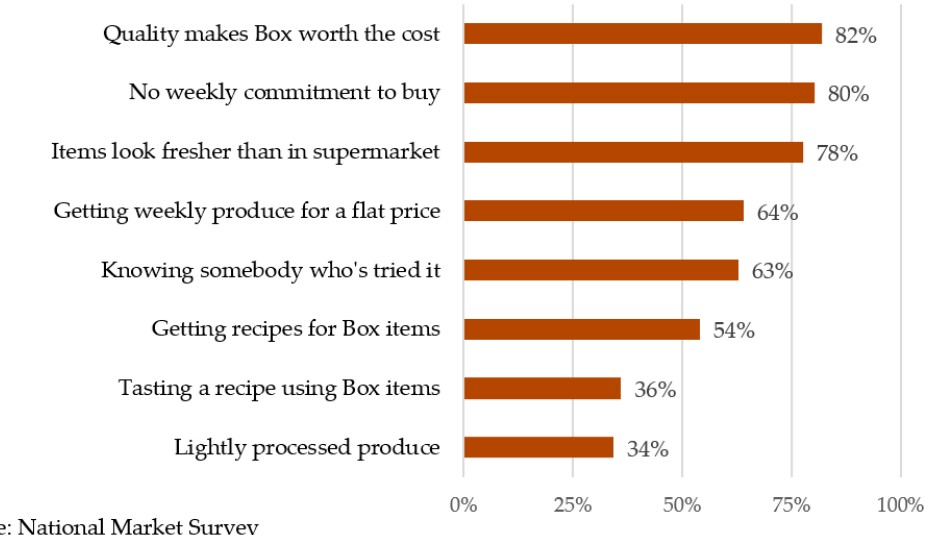

**Figure 3.** Elements increasing likelihood of F3B purchase.

Respondents said they would be less likely to purchase a F3B if they thought the produce would go bad before they could eat it (84%), their family would not eat it (74%), they had to go out of their way to pick it up (72%), or if there was no time to cook or prepare a meal with the produce (65%).

### 4.2. Aim 2: Strategies for Recruitment, Logistics, and Marketing

Local cooperative extension partners first recruited interested farmers, and then found suitable neighboring retailers who were willing to participate. Next, extension convened the farmer and retailer to review basic information about how the F3B model works, and make logistical decisions about how F3B would operate in their setting [83]. The meeting included a frank discussion of what each partner hoped to get from this new arrangement, mutual goals, potential challenges, and requirements for success. A key part of the meeting was choosing the best mode of communication for partners (e.g., telephone, text, or email); previous research noted that establishing regular and timely communication provides the foundation for a strong relationship [80].

During this meeting F3B partners finalized F3B pricing and how customers would pay (cash or credit). To mitigate cost barriers, some farmers priced their F3B lower than neighboring farmers market prices. The retailer usually took payments at time of order, so partners needed to decide how and when payments would be transferred to the farmer, and what the retailer would receive per order (e.g., a percentage of sales, a set dollar amount per week, or other). It should be noted that SNAP/EBT rules precluded the use of this payment method for in-store orders, though pre-ordering through the farmer is allowable.

Partners also reviewed their respective roles and responsibilities. Farmers were clearly responsible for weekly harvesting, packing and delivering boxes, while retailers were responsible for collecting pre-orders, providing orders to the farmer, and receiving boxes. However, they needed to make mutual decisions about the start and end dates for the F3B program, when the weekly items list would be received and posted in the store, the cut-off day for orders to be placed and the count forwarded to the farmer, along with the timing and day for box delivery. Partners also had to decide who would take primary responsibility for marketing the F3B, and tracking and analyzing sales data

Extension provided Farmer-retailer pairs with tailored marketing materials. In 2017, these included exterior signage and interior flyers/whiteboards to advertise changing weekly F3B content offerings; a few retailers displayed an empty F3B box to convey the volume of product being offered. Store VT4A purchased F3B boxes in advance to display for same-day purchase. Another store created advertisements on their gas pumps. In 2018, greater audience reach was achieved through mass direct marketing that sent flyers/postcards to residents within a specific zip code. This method drew the

attention of newspapers, local radio shows, and other farmers, who contacted extension team members for additional details about F3B.

These methods were implemented with varying degrees of fidelity across sites, with two retailers saying in 2017 that advertising would have been more effective if it had been timed to coincide with the availability of the F3B (Stores VT2B and WA2A). Despite the additional efforts in 2018, one retailer felt:

"More people could have learned about it. There could have been a little bit more promotion, maybe in the local paper or something." Store WA3B, 2018

Store WA1A observed that repetition was the key to raising consumer awareness of F3B; still another wondered how to make F3B "stick" in the consumer's consciousness:

"So, the other interesting observation was the almost impossibility of small things—without a huge advertising budgets—Remaining in people's minds long enough for them to develop a pattern." Store CA1A, 2018

Perhaps one of the biggest lessons learned was the importance of discussing how to message F3B's value as it moved down the value chain. Retailers sometimes appeared unfamiliar with the attributes of the farm's brand and were unclear that their role included conveying F3B's embedded values to the customer. This was attributed to inadequate communication between partners, and during the initial meeting to discuss logistics [80]. In addition to having the knowledge and skill to convey embedded values, enthusiasm for the produce was key, according to one farmer:

"But really, it always has a lot to do with store personnel. You know, the store manager, or store personnel, they've gotta be excited about it, or it's just gonna be, like, you know, a sack of potatoes in the backroom for them", "if you get one buyer who's into it, sales really spike up." Farm VT4, 2017

*4.3. Aim 3: Benefits and Challenges for Farmers and Retailers*

4.3.1. Farmer Perceptions

Farmer motivation to try F3B: In post-season interviews, farmers said their primary motive for implementing F3B was to address DTC market saturation and expand their customer base. VT farmers specifically said they were curious about how convenience store shoppers would respond to a box of fresh, local food offered in that setting. F3B also seemed to be a good way to transition to a VBSC at a scale that suited their farm:

"We have a really saturated CSA market, [so] we decided to do the farmer's market. Um, we haven't tried to go into any serious wholesaling, because of the size and the scale of our farm—we're just not big enough to reach that price point, or to even like grow enough to make a living selling wholesale, so, that kind of limits us." Farm VT5, 2018

Additionally, some F3B farmers wanted to further the social goals of AFNs:

"The reason that I want to grow food, and how I envision the food system, is one where people who are farming in a given locality are focused on feeding, first and foremost, the people who live closest to them." Farm WA2, 2017

Trialability played a role, as one WA farmer found it easy to partner with a retailer located along their established travel route, while another said, "It was fun without adding too much, if anything else, to my plate" (Farm WA2, 2017). Similarly, a VT farmer said that already having CSA boxes to deliver made it "very easy to incorporate" delivery of a few extra boxes to the retail site (Farm VT6, 2018). WA farmers acknowledged that getting advice from a trusted extension agent, and knowing retail partners they respected were deciding factors.

In the next sections, we report on how farmers' experience with F3B matched their initial motivations for trying it: profitability, expanding their customer base, and entering into a VBSC arrangement.

Benefits for farmers: As shown in Table 3, F3B sales ranged from $150 to $2720 per site in 2017 (mean = $730) and from $420 to $2490 in 2018 (mean = $1210). Overall, 643 F3B were sold for a total of $14,059 across all sites.

**Table 3.** F3B Sales, 2017 and 2018.

| | 2017 | | 2018 | |
| --- | --- | --- | --- | --- |
| **Site** | **Boxes Sold** | **Total Sales** | **Boxes Sold** | **Total Sales** |
| VT1A | 5 | $150 | — | — |
| VT2B | 23 | $545 | — | — |
| VT4A | 12 | $360 | — | — |
| VT5A | — | — | 25 | $500 |
| VT6A | — | — | 31 | $744 |
| WA1B | 136 | $2720 | 106 | $2120 |
| WA2A | 10 | $200 | 38 | $570 |
| WA3A | 34 | $402 | 60 | $858 |
| CA1A | — | — | 83 | $2490 |
| CA1B | — | — | 66 | $1980 |
| CA1C | — | — | 14 | $420 |
| **Total** | **220** | **$4377** | **423** | **$9682** |

Thus, for most farmers F3B brought only a small increase in revenue in the two years of implementation, due to the small number of boxes sold. Yet farmers understood that it takes time and effort to establish a new market channel, saying:

"When you try to try something new, you expect that you're not going to be efficient the first year, and you're probably not going turn a profit the first year either." Farm VT5, 2018

Despite the modest amount, farmers appreciated gaining additional revenue from F3B sales. They said that implementing F3B helped them reach out to a new audience in a new setting, as they had hoped. As one CA farmer said,

"[It] was additional income for us, so that was beneficial, but also just the potential of creating more relationships, meeting people." Farm CA1, 2018

Farmers saw F3B as a way to attract customers that were either unfamiliar with a CSA-style box or unwilling to commit to purchase for the entire season, thereby opening an untapped market:

"I would glance at the list sometime when I went to drop stuff off", and "I recognized a few of the names, but most of them I didn't, which is kind of indicative to me that there were people who maybe would not have normally gone for local food, organic food, whatever." Farm VT6, 2018

One VT farmer noted that simply changing what the box was called made if more relatable for certain customers:

"Calling it a food box instead of the CSA", "that [term], you know, is probably", "not as familiar to everybody. And then making it accessible in locations that are not sort of your typical CSA locations. I appreciate that effort to just make it, make CSA, a little bit more of an accessible concept and locations." Farm VT1, 2017

Farmers in VT and WA saw value in F3B as a means of boosting consumer awareness of the farm's brand:

"[F3B] was one more feature of our farm that I felt like I could market. And that, therefore, I think, gave us more visibility in the community as a whole" "People love [store 1] and so I think that partnering with them is really good for our image in addition to our sales." Farm VT6, 2018

"Anything that you do that further integrates you into a community—as a business person, as a farmer specifically—then makes your name and your brand more recognizable. That builds relationships, I mean, that does come back to you over time." Farm WA1, 2018

In addition to helping to provide wider awareness and access to the fresh produce they grow, farmers felt that F3B could promote the AFN concept, when marketed correctly:

"So, with the education and the advertisement that could be provided through, you know, the Food Box activity, you know, just drumming it into people's heads that it's here. You know, now's the time. Eat good food. And it's local, and you're supporting a local economy, and you're, you know, making your community, your county, viable in itself and self-sustaining." Farm CA1, 2018

Another farmer expressed confidence that educating consumers and providing them with a quality experience would ultimately lead to profitability for the farm:

"People who hear about us through the [F3B] marketing and then who buy the boxes and get hooked on the food and the experience", "if we can continue to grow the program and get more boxes, that supports us financially", "to be able to build a successful financial business." Farm WA2, 2017

Short VBSC: One farmer saw F3B as mutually advantageous for both producer and retailer. This farmer had previously tried to sell to a local retailer but was unable to strike a deal because the store had limited produce storage and display space. The F3B model provided:

"An opportunity where I could actually have my food distributed through [the store] without inconveniencing them and causing them to change what works for them. And so that was kind of exciting." Farm WA1, 2018.

Another farmer had strong feelings about the mutual benefits of a VBSC model, because it aligned with the AFN values of sustainability and of fairness for producer and retailer:

"I don't think it would be going in the right direction if, for instance, [F3B's model] was just a wholesale price to the grower because the marketer or the retailer has to take the loss, you know? A huge amount of our food is wasted in the market system, and people are kinda paying for it but not totally. And in terms of the energy and life resources that go into producing that food, if it gets wasted, we're wasting our capital, as it were, of the living planet. And we gotta change that, I think." Farm CA1, 2018

As previously reported [80], relationships between retailer, producer and consumer are the cornerstone of VBSC, and farmers recognized that this took time to develop:

"What makes [F3B] successful? A number of things. Significant to this, I would say time spent interacting and kind of a development of a certain level of trust through that", "when you talk about building the community relationships, business relationships, any of those things, it's helpful sometimes just to have like a low-risk get-to-know-you period, and having this as a vehicle for that was probably quite helpful." Farm WA1, 2018

Challenges for farmers: For many farmers, F3B required no additional labor or expense. Most said they simply added F3B to the pack-out/distribution process already in place for their CSA. As one explained,

"You know, when it gets done alongside an existing CSA. It's just part of the numbers and it doesn't feel like it added anything significant." Farm WA3, 2018

Similarly, three WA farmers said that their proximity to the retail pickup site make their delivery routes "easy," and one VT farmer noted:

"The fact that we are already making delivered CSA boxes made this an easy addition. Basically, it was just a few extra delivered boxes that went to a different delivery site. So, it was very easy to incorporate." Farm VT6, 2018

In contrast to the majority, the farmer with three retail partners noted that a significant amount of additional labor was required for deliveries:

"Well, we'd take them down to [Store CA1A] or we'd have to drive out to [Store CA1B], unload those boxes", "so that would be like, you know, another half hour beyond our normal routine, so we're not getting home until 8 or 9 or something." Farm CA1, 2018

This farm had a single CSA delivery site where members packed their own bags, so additional labor was required to assemble F3B boxes. Nonetheless, this farmer believed in the model and felt the benefits outweighed the challenges:

"It's definitely an extra step, but, you know, if it gets vegetables used and creates some income, and, even more importantly, maybe it starts to create outreach and contact, then it's working", "so we're still interested in proceeding with it." Farm CA1, 2018

Scaling up to the next level would have justified hiring additional labor:

"We didn't really get enough customers where we could be like, okay, now we can hire someone to, like, pack the boxes [laughs]. It was just like, oh no, we have to do this thing that we're adding to everything else", "it was a little bit iffy to figure out how to manage that one extra thing." Farm CA1, 2018

Record-keeping was another area where increased effort was required:

"Bookkeeping-wise, it was a real pain for her [the farmer's wife and business partner]. Like, she won't do an entry of a box—it has to be broken down into, systematically. Well, how many pounds of potatoes? How many pounds of squash?" "And, you know, it [F3B] got so successful, she was having to do a lot of direct entries and breaking it down." Farm VT4, 2018

### 4.3.2. Retailer Perceptions

Retailer motivation to try F3B: For the most part, retailers chose to implement F3B in hopes of expanding their customer base:

"I said, 'welcome' [to the F3B model], because anything that you add, especially [to attract] different community, you know, the American versus Hispanic, so I want to get the most customers." Store WA5, 2018

Retailers also wanted to distinguish themselves from competitors by having "something a little bit different that we can offer our customers" (Store WA6, 2018). Some specifically wanted to offer locally grown items:

"We wanted to have local products and we wanted to work with a local farm that was doing it, so we thought [F3B] was a good match." Store VT6, 2018

Like farmers, retailers were motivated by social values:

"Our mission is to help the communities, and that includes creating a local food web. And, one of our goals is to", "[link] the community to local food however we can, whether it's through the café, or through the grocery area, or through classes." Store CA3, 2018

Benefits for retailers: Most retailers reported that F3B had little impact on store profit; five retailers described the financial benefit to their store business as negligible. Most stores sold only a few boxes per week and therefore made only a small amount from the 10% transaction fee and reimbursement of the credit card fees provided by the study.

"we filled, you know, maybe five [boxes] a week or so, so it wasn't a huge, huge deal, but maybe a little bit." Store VT6A, 2018

This retailer thought that F3B sales had potential, and hoped that purchasers would tell their neighbors and friends: "Oh, we did that last year. It was great. You should do it." (Store VT6A, 2018). Other retailers agreed that F3B could be profitable if it were scaled up, though they had difficulty imagining how this would occur since "not very many people that can use [F3B], or they're not gonna drive 5 or 10 min just to come and order that box" (Store WA2B, 2018).

In addition to modest box sales, the hoped-for collateral sales due to increased foot traffic failed to materialize for most retailers. One retailer said it was hard to tell whether F3B purchasers were buying more sundry items than usual since most of them were existing customers (Store WA1B, 2018). Another noted that F3B purchasers did not linger to shop when picking up their box:

"Even at its peak—the dozen people or so that came in to pick up boxes", "mostly they just came and got their boxes and left. A couple of them might have bought an incidental thing or two, but economically overall it was pretty insignificant." Store CA1A, 2018

An unintended consequence for a store that stocked an array of locally grown foods, was that F3B actually detracted from regular produce sales.

"It's possible that our participation in the food box cut into our other produce sales because the customers who were interested in buying local produce at the store decided that they would get the food box because it was a great program and a great value." "They weren't spending that money on the produce that we were buying wholesale from other farmers, which we actually did make some profit on." Store VT5A, 2018

Another retailer noted that while F3B was a "really great value" that gave consumers high-quality local produce at prices below the farmers' market, this was not a profitable model for the store:

"I think the missing piece is that it doesn't really make sense [for the store owner]. If the sole interest of the business is to be profitable, then the food box program doesn't quite serve that goal." Store VT5A, 2018

This retailer thought that a wholesale model would have garnered more profit for the store:

"If we had purchased all of that produce at a wholesale place from [the partner farm] and then marked it up and sold it retail, the store would've made more profit", "I guess a businessperson might look at [F3B] and be like, 'Well, what's in it for me?'" Store VT5A, 2018

Challenges for Retailers: Many retailers said that implementing F3B had few challenges, and required little additional expense or staff time. However, the F3B extension team observed that the participating retail stores were often high-paced operations whose small staff had to manage many details [83]. Introducing F3B, with its specific ordering and pickup protocols, constituted yet another procedure for staff to remember. If the retailer had high staff turnover, this was even more challenging because the owner had to take time to train new staff in how to post weekly box contents, manage transaction logistics, tally orders for the week, and notify the farmer of the weekly box totals.

### 4.3.3. Facilitating Factors, According to Farmers and Retailers

Farm WA1 attributed their success to the retailer's authentic relationship with customers as well as a country store ambiance that was conducive to buying fresh local foods, in contrast to an overstimulating convenience store.

"I think a lot of the credit would go towards them [the retailer] and just the people they are, and the way they're able to structure and operate their business, and the people that they have to run it for them." Farm WA1, 2017

Farmers and retailers had lingering questions about how to best reach various consumer subgroups. Initially, F3B was intended to attract consumers who had never participated in CSA, recapture those who had left the CSA for financial or other reasons, and entice farmers market patrons looking for better prices or more convenient shopping times and locations [84].

While farmers like WA1 saw country stores as ideal setting to find rural food enthusiasts who might be interested in F3B, they were uncertain about how to reach that untapped pool of consumers they described as, "people who don't know they really want that stuff" (Farm VT4, 2017). They agreed that convenience stores, where most F3B implementations took place, had clientele who typically seek local produce. They also noted that these stores were often in walkable, centralized locations, which addressed potential transportation and convenience barriers for shoppers (Farm VT4, 2017). Yet farmers wondered how to interest customers who had other priorities when shopping at convenience stores [22].

Store VT5A described modifications to their store environment that, over time, encouraged typical convenience store shoppers to try locally-grown products, including F3B. Previous store owners targeted a working-class demographic who typically sought chips, soda, beer, and lottery tickets. VT5A owners introduced locally-grown produce, eggs and bread to attract a new market, but kept offering typical convenience-store fare to retain existing customers and "make it a place for everybody", "a very inclusive and accessible place" (Store VT5A, 2018). They were largely successful in attracting both kinds of customer, and found that over time"

"Someone who had been coming to the store for years maybe and just buying a bag of potato chips and a soda would suddenly discover that we had these incredible baguettes that were made by a French guy. And then they would start buying those on a regular basis. And I think the same thing applied to the food box where we were really surprised at who actually was interested in getting the food box on a weekly basis, and it wasn't just the people that you might guess." Store VT5A, 2018.

### 4.4. Aim 4: Consumer Attitudes and Purchase Behavior

Of the 607 boxes sold through 13 retail sites over two years, 59 F3B customers returned a completed purchaser survey. Most respondents described themselves as non-Hispanic white, and female (98 and 91 percent, respectively). Compared to the area demographics of F3B implementation communities (Table 2), respondents tended to be older (mean age, 63 years) and affluent, with 39 percent having incomes over $75,000 [23].

Figure 4 below displays respondent experience with F3B, based on their Likert scale agreement with various statements. As in the F3B national market survey, many purchasers were motivated by an interest in supporting local farmers. Regarding diffusion of innovation indicators, F3B was easy to try (trialability) and was compatible with purchasers' usual eating habits, though less so with their meal planning and shopping habits. Less than half the respondents felt that F3B had a relative price advantage over supermarkets, though nearly all said they could afford to purchase it. Most respondents agreed that they could prepare the items found in their box, saw how F3B could benefit the store, and hoped the store would carry F3B next year.

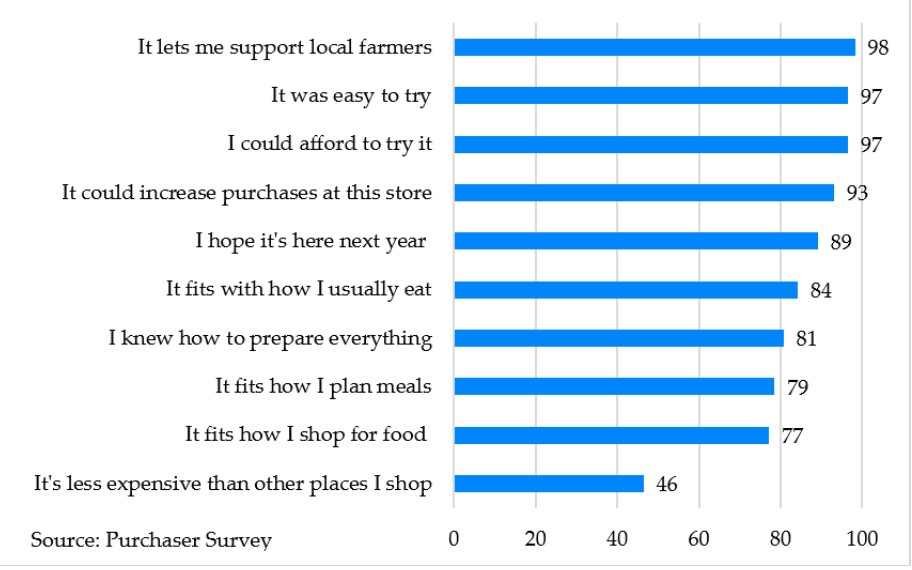

**Figure 4.** Purchaser Experience of F3B.

Respondents were generally satisfied with F3B features, particularly the quality and variety of the produce (97 and 86 percent, respectively). "Quality," referring to the perceived freshness, taste, and appearance of the produce, was often mentioned in open-ended questions about what purchasers liked best about F3B, with one commenting on "all the beautiful and delicious vegetables," and another mentioning "top quality, clean produce."

These reactions were affirmed in post-season interviews with retailers, who observed that F3B purchasers generally felt the produce was "always really good quality", "a lot of people commented on that, how good it was" (Store WA2, 2018). Retailers also commented on customers' delight with the variety of produce:

"Oh, they were very happy. I mean, you know, first time they went like, 'Oh, that's what I get? Great!' And they were just excited with each time that was different produce." (Store VT6, 2018)

Some respondents mentioned specific items they enjoyed finding in their F3B (e.g., fresh carrots, cantaloupe), while others mentioned items they did not like (e.g., peppers, eggplant). Respondents had varied reactions towards uncommon vegetables. Some approached novelty with a spirit of adventure and "rose to the challenge of eating everything received," while others wanted more help to identify and cook unfamiliar items.

Most purchasers expressed satisfaction with the logistical aspects of F3B: ease of ordering, retailer interactions, and pickup convenience (93, 92 and 89 percent, respectively). Many respondents said they appreciated the freedom of ordering on a week-to-week basis, while still being able to support local farms. As one respondent said, "I liked being able to continue supporting [farm], since I had previously been a CSA member but couldn't travel to the farm this year to get the food."

However, a few purchasers felt inconvenienced due to what they perceived as poor organization and communication between farm and store. As one respondent said, "The weekly system wasn't consistent in terms of emails from the farm or location to get the food and timing in the store." Another said F3B was "not as convenient to pick up as shopping at the grocery store," without further elaboration.

About three-quarters of respondents indicated satisfaction that F3B provided value for the money. Yet quantity was an issue for some, with several respondents mentioning that for the price, "there was not much or not enough of the same thing." Perceptions about quantity were subjective; for example, one respondent said the amount was "just enough for a week for 2 people," while another said the box "would not serve more than a 2-3 [member] family if all items were combined." Overall, most respondents were satisfied with both the quantity and variety F3B gave them, saying they got "all the veggies I want, 0 wasted, 'weird' CSA veggies".

Open-ended responses revealed respondents' desire for more information about F3B items, including storage instructions and recipes. This is important to note, as recipes and information about item contents could have supported self-efficacy to use and enjoy box contents, and also serve as an opportunity to educate F3B purchasers about the farm, embedded values of the product and principles of AFNs, as previously mentioned by Farm CA1.

Perceived value may have influenced repeat purchases. One retailer noted that repeat business came from customers that were really enthused about F3B, while others "looked at it and thought, Well, that is not 30 bucks worth of produce!" (Store CA1, 2018)

## 5. Discussion

The original aims of the research component of the F3B project were to assess F3B market potential in three geographically diverse and rural areas; determine strategies for recruitment and logistics; describe F3B benefits and challenges for farmers and retailers, and measure consumer response in terms of attitudes and purchase behavior.

### 5.1. F3B Market Potential

F3B was trialed in three states, one in the northeast and two in the west. This gave us an opportunity to test the model in areas of the country with different degrees of market saturation: according to the F3B national market survey, 90% of shoppers in the Northeast had purchased food through DTC venues, compared to 64% of shoppers in the West. F3B implementation sites were in mostly rural or peri-urban communities with high car dependence. Some F3B sites had nearby grocery stores, a potential source of competition but also indicative of food access. Other implementation sites had no proximal grocery stores, and few had neighboring DTC venues.

We found that while F3B provided only nominal revenue due to modest box sales, farmers were satisfied with other, non-monetary benefits. These included reaching a new market segment, expanding their customer base, promoting the farm's brand, and developing a mutually beneficial partnership with retailers. While most farmers said F3B was easily incorporated into their existing packing and delivery routines, some reported that record-keeping required extra time. Establishing a working relationship with retail partners also took time and effort, though few farmers said this constituted a significant burden. For most farmers, F3B marked the first step towards intermediated direct sales. Previous research has demonstrated that farmers selling through a mixture of DTC and intermediated channels bring in higher revenues than those selling through DTC alone [44].

For retailers, F3B generated fewer collateral sales than anticipated. Yet business owners noted minimal burden with implementing F3B. Though F3B contributed little to the profitability of their

business overall, retailers found value in being able to offer a new product (local foods) to their customers without investment in perishable stock and equipment.

## 5.2. Recruitment Logistics, and Marketing

Retailer location was a critical component of successful F3B implementation. Most retail sites were conveniently situated along consumers' regular travel routes; thus, farmers did not have to travel too far outside their normal delivery routes. The best retail-farm matches were ones in which both partners were motivated to put time and effort into piloting F3B, and where farmer and retailer understood their respective roles. Relationship skills were needed to establish good communication habits, which often began during a joint meeting, facilitated by extension staff, to decide arrangements for transmitting weekly orders and scheduling deliveries. Educating F3B retailers on how to convey F3B's value by promoting the farm's distinctive brand was a potential area for improvement. This and other lessons learned by extension in the course of providing technical assistance have been assembled into a virtual short course for farmers, retailers, extension and others on how to start an F3B [84].

## 5.3. Benefits and Challenges For Farmers And Retailers

Table 4 displays the baseline challenges for farmers, retailers and rural consumers in rural communities; our assumptions about how F3B would address each challenge; and post-implementation insights on the extent to which each challenge was addressed as anticipated.

**Table 4.** Challenges, Potential Benefits and Findings.

|  | | Beneficial Effects of F3B | |
|---|---|---|---|
|  | **Challenges** | **Initial Assumptions** | **Post-Implementation Insights** |
| **Farmers** | Hard to supply wholesalers due to farmer's smaller farming operation & crop diversification. | F3B adjusts to what the farmer can produce, allowing week-to-week flexibility in the items provided. | Degree to which F3B helped move surplus product remains unclear. |
|  | Farmers have little to no control over wholesale pricing. | Farmers can set prices & quantities for F3B, as appropriate. | Farmers set prices and quantities to suit their business, depending on what the market can bear. |
|  | CSA & farmers market segments may be saturated. | Reaches those who are unwilling or unable to purchase local food through a CSA or farmers market. | Availability of online ordering and acceptance of SNAP may be needed to attract additional customers. |
|  | Time to drive and sell at farmers' may not result in sale of all produce, leading to waste. | The farmer can drop off the F3B rather than spend time staffing a booth. Pre-ordering ensures produce will be sold. | Taking advantage of usual delivery route works best for farmers. Advantages offset by time farmer must spend communicating & building relationship with the retail partner. |
|  | To supply institutional buyers, production would have to increase to have sufficient volume. | F3B model allows farmers to bring in revenue while they expand production to diversify their business to include sales to institutions. | Establishing the F3B model as a new part of the farm business takes time and effort, leaving less time to increase production. |
| **Retailers** | Small retail businesses may falter due to lack of foot traffic & sales to local residents. | F3B pickup gives consumers a reason to visit retailers & may lead to added sales. Retailers get a small percentage of sale and transaction fees (e.g., credit card fees). | Online ordering is more convenient for consumers, who may not like making 2 trips to order/pickup. Pickup still brings consumers into the store. |
|  | Perishable produce not stocked due to lack of cold storage space & insufficient sales volume. | Retailers do not have to invest in perishable stock or equipment, since they do not have to pay for F3B or store them. | Lower than expected sales volume and short period of time that boxes where in-store did not negatively impact store's space. |

**Table 4.** *Cont.*

| | | Beneficial Effects of F3B | |
| | Challenges | Initial Assumptions | Post-Implementation Insights |
|---|---|---|---|
| **Consumers** | Up-front CSA costs are a barrier for those with low income. | F3B have lower up-front cost since they are purchased week to week, & no on-going commitment. | Few low-income people bought F3B. Thus, barriers remain, such as ability to use SNAP/EBT to pre- boxes from retailers. |
| | Food prices at farmers' market may be prohibitively high, or perceived to be too expensive. | F3B are priced for affordability relative to seasonal CSA prices. | F3B pricing typically depends on the minimum profit acceptable to the farmer, which may not be much less than farmers market or supermarket prices. |
| | Rural residents may lack transportation to CSA site, farmers market or food store. | Retail site is located in the town center, along consumers' usual travel routes. | Finding a good match, including a retail site with an accessible location, is a key success factor for F3B. |
| | Consumers may perceive Farmer's Market as an elite social space. | Retail site is a familiar & acceptable place for consumer. | Understanding convenience store shoppers' needs and expectations is key to encourage them to purchase a F3B. |
| | Farmers markets & CSAs have limited days or hours of operation. | Retail sites are open daily, & offer longer hours for pickup than a typical CSA. | Most F3B boxes were available at stores within 24 h. Previous CSA members were more satisfied with F3B pickup than those who had never bought a CSA. |
| | Customers like to shop at larger supermarkets to buy additional groceries with their produce. | Customers are able to purchase other items such as milk, eggs, & bread when picking up a Food Box. | Few reported collateral sales in the purchaser survey. |

F3B was designed to address DTC market saturation by opening new markets for farmers, without the staffing time required by other DTC models or risk of unsold product. Most of our assumptions about F3B's benefits were confirmed in post-intervention findings, though time saved by not staffing a farmers' market stall may have been offset by the time required to build a relationship with the retailer. Because the farmer retained ownership of the box until a sale was made, they were able to maintain control over its contents, volume, and pricing. As with other VBSCs, F3B introduced an additional step in the supply chain between farmer and customer, meaning that the farmer had to rely on the retailers' efforts to pre-sell the boxes and was not able to directly build relationships with customers.

Initially, stores used outdoor sandwich boards with updated in-store flyers and whiteboards to advertise weekly F3B content offerings. In the second year, postcard advertising and newspaper and television stories were used to bring F3B to the attention of a wider audience. However, these simple methods were not implemented consistently across sites. Further, farmers remained perplexed over what it would take to attract typical convenience store shoppers to F3B given that they likely differ from usual DTC customers in their values and expectations regarding price and convenience.

Within the retail space, F3B sales may have been limited by the need for in-person ordering. which required customers to make two separate visits to complete a transaction. Moving to an online ordering system would diminish potential foot traffic associated with two visits, but the increased convenience could attract a wider array of customers. Similarly, exploring solutions to allow the use of SNAP benefits to purchase F3B might overcome a barrier for low-income consumers.

*5.4. Consumer Attitudes and Purchase Behavior*

Most F3B purchasers expressed satisfaction with the quality, quantity and variety of produce in their F3B each week. A high proportion considered F3B to be a good value for the money, and half said the F3B price compared favorably with other places they shop for produce. While a few suggested that farmers either increase the quantity of F3B produce or lower the price, though this might not be

economically feasible for the farmers. Wanting food to be both inexpensive and sustainably produced reflects contradictory consumer expectations, as previously described [23,85]. To accommodate shoppers with contradictory perceptions about whether the amount of food in the weekly F3B is "enough", farmers might offer various F3B sizes to meet the needs of different households, thereby increasing consumer satisfaction without exploiting the farmer.

Compared with other DTC options, F3B compared favorably. Purchasers appreciated the ability to order F3B on a week-to-week basis, without making a long-term commitment. Most of the inconveniences mentioned by purchasers were attributable to issues at the farm and store level rather than to problems with the F3B model itself.

F3B purchasers said they wanted more information to help identify and prepare items in the weekly F3B; this could provide an opportunity to educate consumers about AFN values in general, and highlight differentiators associated with each farm's brand. Research suggests that if information that supports the value of regionally produced food is properly conveyed, consumers will be more likely to purchase and overlook inconveniences [86]. This aligns with Farm CA1's recommendation to educate purchasers about AFN, particularly for those new to the concept.

### 5.5. F3B's Relevance for Emerging Food System Problems

The DTC market landscape has changed significantly in the face of the COVID-19 pandemic, revealing the fragility of the conventional supply chain, raising concerns about the safety of foods packed and processed in crowded, centralized facilities, and the reliability of food stocks [87–89]. Sheltering in place has altered food-purchase patterns, with an uptick in CSA subscriptions, online ordering and requests for home delivery. People are cooking more, and willing to experiment with new recipes and foods. Additionally, more people have experienced a mixed produce box, due to the Farmers to Families Food Box program funded through the CARES Act, which addresses farmers' loss of institutional contracts from schools and restaurants, along with the increased need for food assistance due to COVID-related unemployment. Interestingly, it is the small and medium farmers and values-based supply chains that have demonstrated an ability to pivot quickly in response to the shock [87].

### 5.6. Limitations

Though F3B was originally intended to address the lack of access to fresh produce in rural areas, this project did not attempt to measure its impact on food access and food security at the population level, due to the modest number of boxes sold at each site. Similarly, due to the relatively short duration of this trial, we tracked the number of boxes sold, but not farmer revenue, collateral sales to retailers, or impacts on the local economy. Finally, our approach to recruitment and logistics was to explore what worked best for each farmer-retailer pair within their local context, rather than attempt to determine the best practices for success, though extension compiled a set of lessons learned across implementation sites, with tips on how to address common problems when starting a F3B program [84].

F3B received generally positive feedback from purchasers, with few complaints. Yet since fewer than 10% of boxes sold resulted in response to the purchaser survey, it is likely our results reflect only the most satisfied customers. Attempts on the part of extension to solicit opinions from those who did not purchase F3B failed to produce usable data. Thus, the information we have is insufficient to explain causes for low box sales and generate ideas for improvement.

### 5.7. Implications for Future Research and Practice

We piloted F3B thinking that it would appeal to those who wanted to buy local produce, but with convenience and freedom from long-term commitment. This was borne out somewhat in the market survey and in purchaser feedback, though box sales were modest. Growing the F3B model in rural retail venues will require a better understanding of the values, attitudes, and behaviors of rural consumers regarding local foods, particularly of convenience store shoppers, which may differ from the typical

AFN enthusiast. A related question for research is whether rural consumers respond to advertising that emphasizes embedded AFN values of local foods, as other research has found. Findings from research on rural consumers could then be incorporated into F3B marketing plans to further develop the community- and in-store advertising used in this pilot. An improved marketing plan could also specify how the retailer will convey the farm's brand, an essential feature of VBSC. Extension would be an ideal partner to provide technical assistance for market plan development that incorporates new research findings on rural consumers.

Low sales may also have been due to the need for purchasers to visit the store twice to order and pick up the box. The inability of the retailer to accept SNAP benefits to pay for F3B may have limited sales to low-income customers. Retailers intending to adopt the F3B model will need to find ways to address those issues; they may also wish to provide flexible options, such as various box sizes, lightly processed vegetables and specialty product offerings. The F3B Short Course and Toolkit provide evidence- and practice-based guidance that can help new farmer-retailer pairs start their own F3B project; extension is an ideal partner to help them do this, as well as help them explore strategies to address the challenges described above.

## 6. Conclusions

Findings from the F3B project demonstrate proof of concept. This study offers useful information about the market potential of a short VBSC innovation designed to expand markets for CSA farmers, provide a low-risk way for retailers to offer locally grown foods to their customers, and increase access to fresh produce for rural consumers.

Despite low initial profitability, retailers and farmers found F3B to be a worthwhile endeavor, enhancing their brand and increasing the visibility of their businesses. Though there is risk associated with investing time and labor in business innovation that will not see a return until the product "takes off", participating F3B farmers and retailers expressed a belief in F3B's potential, and seemed willing to give it time to take hold. Though sales were modest, those who did purchase it voiced satisfaction with F3B, and the majority hoped it would be offered in the following year.

Future research should focus on investigating the attitudes and needs of rural consumers regarding local foods, as well as which advertising strategies are most appealing and result in sales. As the F3B model is more widely established, it will be possible to conduct economic studies on the financial benefits of local economies and to farm and rural retail businesses, as well as estimating the degree to which F3B addresses lack of food access and food security in rural areas.

The current environment presents both opportunities and challenges for the F3B model [87]. F3B's appeal is now enhanced by the fact that its contents are both traceable and safer, with fewer hands touching the produce in the box. Showcasing those virtues depend on the ability of the retailer to convey the farm's brand, to increase trust in the farmer and the food they produce, as well as build confidence that this short VBSC can be counted on to deliver healthy foods if the conventional food system supply chains falter. Barriers to purchase—such as inability to accept SNAP/EBT, lack of an online ordering system, limited sizes and product offerings—must be resolved in order for the model to be more competitive with traditional supermarkets that offer those amenities.

**Author Contributions:** Conceptualization, M.S. Methodology, M.S., W.W., Z.V.V., L.G. and J.K. Formal Analysis, W.W., M.S., Z.V.V., L.G. and J.K. Investigation, W.W., L.G., D.S., H.E. and J.V.S.K. Writing (Original Draft Preparation), M.S. Writing (Review & Editing), M.S., W.W., D.S., L.C.C., J.V.S.K. and J.K. Visualization, M.S. and W.W. Supervision, L.C.C. and J.K. Project Administration, W.W. and L.G. Funding Acquisition, J.K., L.C.C. and M.S. All authors have read and agreed to the published version of the manuscript.

**Funding:** This work is supported by Innovation for Rural Entrepreneurs and Communities project no. VT-0075CG from the USDA National Institute of Food and Agriculture.

**Conflicts of Interest:** The authors declare no conflict of interest.

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
