# Peer review of "Evaluation of Farm Fresh Food Boxes: A Hybrid Alternative Food Network Market Innovation"

_sustainability, doi:10.3390/su122410406_

Round 1

Reviewer 1 Report

The analyzed topic is very interesting and very up to date. 

The aims of the paper were clearly stated. However hypothesis were not described. It is hard to refer to an article when one does not know the purpose and assumed research hypothesis. 

In my opinion the title is adequate to the research problem being undertaken. The article has been correctly divided into relevant sections, and their content coincides with their titles. However, individual sections showed the studied phenomenon very briefly. This especially applies to the introduction and conclusions section.The presented research is very broad and presents the problem in an interesting way. A great effort put into obtaining the research results. However, in my opinion, the description of the sample for research needs to be extended. Lack of information about what kind of research were conducted (interviews? Surveys?), On what sample (how many people? Companies? What percentage of the total population?) Who conducted the research?  

The correct terminology was used. The language of the article is mature, correct, adequate.

A satisfactory number of references were used (over 90). A solid theoretical basis for work was prepared. All references seems to be appropriate and adequate to the topic and to related and previous work. I don't detect inappropriate self-citations by authors. I have found that about 16 of the featured literature publications are from the authors of this article. However, considering the fact, that the article has 9 authors, this is not an incorrect number. It shows that the authors have experience in the presented topic.

I don't understand how Authors wrote the part of the article entitled: "Marketing F3B to Convenience Store Shoppers"Which part of the article does this item belong to?As for the aesthetics of the article, the first half of paper was very aesthetic. Later (from the results section) it looks like it is glued together from the working items. I think it is worth improving the aesthetics of the paper.I can't find the Figure 2 in the paper....The font size and type in Figures 3 and 4 are not adapted to the size and type of the font in the paper.It is not always possible to find references to the presented Figures in the text, e.g. to Figure 3 

I believe that the source of each table and figures should be indicated. The Authors did not indicate data sources in tables and figures. 

Footnotes and references are generally correctly formulated.

However: Paper numbers in the References section should not be in parentheses, but should be followed by a dot. Instead of: (1), it should be: 1.The last paragraph before Conclusions section: footnote 89 should be entered with the same font size as the rest of the text.

Author Response

Reviewer1:

Thank you for taking the time to review this paper and provide valuable feedback. We appreciate your kind words and have addressed the concerns you raise in the following ways:

  • Hypothesis: As this is an evaluation of a pilot study, that relies mainly on qualitative data, we did not state a formal hypothesis or do hypothesis testing as part of our analysis. However, we have revised the methods section to include a description of the study design to make this more clear.
  • Methods We have modified this section to include a description of the design as a mixed methods research study that uses quantitative and qualitative data from primary and secondary sources to assess each research aim. We also described the constructs, variables, data sources and analytic methods and describe how data are used to assess each research aim. We described primary data were collected, including a description of survey tools and interview guides. We also note the sample size and describe the population from which the sample was drawn. For example, we compared the demographics of the respondents to the National Marketing survey to that of the US population.
  • Marketing section We originally reported “Marketing F3B to Convenience Store Shoppers” under the section on Consumer response to F3B (Aim 4). But, based on your comments, we now see this is confusing and not a good fit. The material has been rearranged to report some information under Aim 2 (logistics) and some in a new section on "Facilitating factors, according to farmers and retailers" under Aim 3.
  • Figures: We edited our figures to make sure the font matched the text in terms of type and size. We also included the data sources on each figure. Figure 2 was inadvertently dropped from the manuscript during the process of uploading to MDPI. This has been corrected. We included the data source on each figure.
  • Aesthetics: We reviewed the manuscript to improve the flow of the results section to mitigate the sense that this section was “patched together”. We carefully examined the language make sure this section was concise, clear and non-repetitive. As mentioned above, we rearranged material on marketing that originally appeared under Consumer response to F3B (Aim 4) and reported it as part of the results for determining recruitment and logistics (Aim 2) and benefits for farmers and retailers (Aim 3), which was a better fit.
  • Reference Section: Finally, we corrected the numbers in the reference section by removing the parenthesis in from of each: g., 1. Instead of (1).
  • Note, we have also made revisions to the discussion section to include limitations of this research, and the implications for future research and practice.

Thank you for your input. We believe these revisions will strengthen our paper.

Reviewer 2 Report

I think the paper is rather well described and interesting one.

SI suggest followings improvements:

-The conclusion is short and insufficient. There should be more information about the results of the research,

-It's worth to describe research limitation in the text.

-It's good to describe the social and scientific implication of the reserach in the conclusion.

I think the aims of the text should be also in abstract.

Author Response

Thank you for taking the time to review this paper and provide valuable feedback. We appreciate your kind words and have addressed the concerns you raise in the following ways:

  • Research limitations We now address the limitations of this study in the discussion
  • Research implications: We now describe how this research will impact current market practices for farmers and retailers, and we discuss topics for future research.
  • Abstract: We included the research aims in the abstract, and also explicitly noted key research findings in the abstract.
  • Conclusion: The discussion provided an in depth summary of research findings, which we did not wish to repeat in the conclusions section. However, we have now included a brief description of the main highlights of the research in the conclusions sections, and also mention how this will affect future farmer-retailer enterprises and marketing efforts for local foods, as well as future research on this topic.

Thank you for your input. We believe these revisions will strengthen our paper.

Reviewer 3 Report

The study aims at investigating the " Evaluation of Farm Fresh Food Boxes, a hybrid alternative food network market innovation".
Although the paper deals with an interesting topic, major revisions are required.
The paper is not satisfactory written, needs a careful editing.
However it is recommended:
- The study aim and background are not well presented, repetitions occurring in the paper should be avoided.
- Introduction and, above all, the conclusions can be improved in order to show better aim and results for further studies in the topic.
- The methodological framework is not linear and have particular problems or omissions allowing you to make everything work strictly from a scientific perspective.
- Carefully check the full text. Especially regarding the state of the art.
- Please do more to highlight how the work advances or increments the field from the present state of knowledge and provide a clear justification for your work.
- Conclusion section needs improvement. Please provide more quantitative key contributions of the study with proper discussions, highlight the limitations of this study and the future work.
- English proofreading is needed.
- Literature survey is not sufficient.
- Authors to take into account some recent references.

Author Response

Thank you for taking the time to review this paper and provide valuable feedback. We appreciate your kind words and have addressed the concerns you raise in the following ways

  • Methods We have modified this section to include a description of the design as a mixed methods research study that uses quantitative and qualitative data from primary and secondary sources to assess each research aim. We also described the constructs, variables, data sources and analytic methods and describe how data are used to assess each research aim. We described primary data were collected, including a description of survey tools and interview guides. We also note the sample size and describe the population from which the sample was drawn. For example, we compared the demographics of the respondents to the National Marketing survey to that of the US population.
  • Research limitations We now address the limitations of this study in the discussion
  • Implications for future research and practice: We now describe how this research will impact current market practices for farmers and retailers, and we discuss topics for future research.
  • Proofreading: We conducted another close reading to check grammar, spelling, and punctuation to ensure there were no outstanding errors. We removed redundancies and tried to make the text more clear and concise.
  • Literature review: We feel that our references represent the most current literature available on this topic. 40% of the literature cited was published within the last 5 years. Our references came from two o reviews of the literature on food box programs and on consumer attitudes and behaviors regarding locally grown foods that two of the authors recently completed.

Thank you for your input. We believe these revisions will strengthen our paper.

Reviewer 4 Report

The authors work on an important topic. The paper is well-written. The only issue needs to be addressed:

The sample sizes and measurement items of the surveys and interviews should be indicated in the manuscript.

Author Response

Thank you for taking the time to review this paper and provide valuable feedback. We appreciate your kind words and have addressed the concerns you raise in the following ways:

  • Methods We have modified the methods section to identify this as a mixed methods research study that uses quantitative and qualitative data from primary and secondary sources to assess four research aims. We also describe the constructs, variables and data sources used to address each research aim. We describe how primary data were collected, noting the sample size and describing the population from which the sample was drawn. We describe the survey tools and interview guides, and how data collected by using these tools were analyzed.

Thank you for your input. We believe these revisions will strengthen our paper.

Round 2

Reviewer 3 Report

Dear Authors,

I am happy to inform you that I have accepted your revision of the manuscript and will recommend it for publication without further changes. Congratulations. I look forward to reading it online.

Thank you for the opportunity to let me contribute a small part to your publication.